# Why do inverse models disagree? A case study with two European CO$_2$ inversions

Saqr Munassar[1,2], Guillaume Monteil[3], Marko Scholze[3], Ute Karstens[4], Christian Rödenbeck[1], Frank-Thomas Koch[1,5], Kai U. Totsche[6], and Christoph Gerbig[1]

[1]Department of Biogeochemical Signals, Max-Planck Institute for Biogeochemistry, Jena, Germany
[2]Department of Physics, Faculty of Sciences, Ibb University, Ibb, Yemen
[3]Department of Physical Geography and Ecosystem Science, Lund University, Lund, Sweden
[4]ICOS Carbon Portal at Lund University, Lund, Sweden
[5]Meteorological Observatory Hohenpeissenberg, Deutscher Wetterdienst, Hohenpeißenberg, Germany
[6]Institute of Geoscience, Friedrich Schiller University, Jena, Germany

*Correspondence to*: Saqr Munassar (smunas@bgc-jena.mpg.de)

**Abstract.** We present an analysis of atmospheric transport impact on estimating CO$_2$ fluxes using two atmospheric inversion systems (CarboScope Regional (CSR) and LUMIA) over Europe in 2018. The main focus of this study is to quantify the dominant drivers of spread amid CO$_2$ estimates derived from atmospheric tracer inversions. The Lagrangian transport models STILT and FLEXPART were used to assess the impact of mesoscale transport. The impact of lateral boundary conditions for CO$_2$ was assessed by using two different estimates, from the global inversion systems CarboScope (TM3) and TM5-4DVAR. CO$_2$ estimates calculated with an ensemble of eight inversions differing in the regional and global transport models, as well as the inversion systems show a relatively large spread for the annual fluxes, ranging between -0.72 and 0.20 PgC yr$^{-1}$, larger than the prior uncertainty of 0.47 PgC yr$^{-1}$. The discrepancies in annual budget are primarily caused by differences in the mesoscale transport model (0.51 PgC yr$^{-1}$), in comparison with 0.23 and 0.10 (PgC yr$^{-1}$) that resulted from the far-field contributions and the inversion systems, respectively. Additionally, varying the mesoscale transport caused large discrepancies in spatial and temporal patterns, while changing the lateral boundary conditions lead to more homogeneous spatial and temporal impact. We further investigated the origin of the discrepancies between transport models. The meteorological forcing parameters (forecasts versus reanalysis obtained from ECMWF data products) used to drive the transport models are responsible for a small part of the differences in CO$_2$ estimates, but the largest impact seems to come from the transport model schemes. Although a good convergence in the differences between the inversion systems was achieved by applying a strict protocol of using identical priors, and atmospheric datasets, there was a non-negligible impact arising from applying a different inversion system. Specifically, the choice of prior error structure accounted for a large part of system-to-system differences.

# 1    Introduction

Inverse modeling has been increasingly used to infer surface-atmosphere fluxes of carbon dioxide ($CO_2$), from observations of dry mole fractions made at spatiotemporal points across an observational network (Enting and Newsam, 1990; Bousquet et al., 1999). Reducing uncertainty in the flux estimates is, therefore, essential to reliably quantify the carbon budget (Friedlingstein et al., 2022; Le Quéré et al., 2018) as well as to improve our understanding about the variability and trends of the carbon cycle over times at finer regional scales, in particular in response to the climate perturbation caused by the increase of anthropogenic emissions (Shi et al., 2021). The estimates obtained from atmospheric tracer inversions still demonstrate large deviations due to manifold sources of uncertainty such as using different data, inversion schemes, and atmospheric transport models (Baker et al., 2006; Gurney et al., 2016), either at global scales or, to a larger extent, at regional scales. Although the global inversions can provide convergent estimations of the global carbon budgets, they are limited by the coarse resolution of atmospheric transport that may not allow for a realistic representation of the observations at complex mesoscale terrains. In turn, performing regional inversions with mesoscale transport models has offered a better opportunity to represent and make use of the dense measurements available at all the sites across regional domains (Broquet et al., 2013; Kountouris et al., 2018a; Lauvaux et al., 2016), specifically after the expanding coverage of data over large areas in the recent years as has been established, for example, over Europe by the Integrated Carbon Observation System (ICOS). Although $CO_2$ fluxes constrained by atmospheric data in the Bayesian inversion framework inherit a dominant spatial and temporal pattern from the atmospheric signal, the a-posteriori fluxes still suffer from a large spread when using different global and mesoscale transport models (Rivier et al., 2010).

As a first intercomparison between six regional inversions covering a wide range of system characteristics –e.g., prior fluxes, inversion approaches, and transport models, the EUROCOM experiment (Monteil et al., 2020) suggested large spreads in posterior estimates over Europe, particularly over regions that are poorly constrained by atmospheric data. This, on the one hand, partly indicates the sensitivity of the a-posteriori estimates to the observations and to the a-priori models as explained in Munassar et al. (2022). On the other hand, inaccuracies in atmospheric transport (Schuh et al., 2019), far-field contributions, and the configurations of inversions are responsible for part of that spread. A further study suggests that uncertainties in both transport and $CO_2$ fluxes contribute equally to the uncertainties in $CO_2$ dry mole fraction simulations, displaying similar temporal and spatial patterns (Chen et al., 2019).

The atmospheric transport relates the measured tracer concentration to its possible sources and sinks, which are adjusted in order to fit the modelled concentrations to observed data. However, inaccuracies in representing the real atmospheric dynamics by transport models lead to uncertainties in $CO_2$ flux estimates. This kind of errors can emerge from both simplified parameterizations of real physics and model parameters themselves (Engelen, 2002). The atmospheric transport models rely on a mesoscale representation of air mass movements, which cannot completely reproduce the observed fine scale variability of tracer concentration, leading to the so-called representation error. As a result, inversions cannot solve for fluxes at a lower spatial and temporal resolutions than that of their transport model resulting in aggregation errors (Kaminski

et al., 2001). Additionally, atmospheric transport models are typically driven by meteorological data available from operational weather forecast models or reanalysis data optimised against observations and dynamical model forecasts. However, such meteorological fields have uncertainties owing to errors and gaps in the observations and errors in the weather forecast models (Deng et al., 2017; Liu et al., 2011; Tolk et al., 2008).

As the lateral boundaries are provided from a global model run at lower resolution than the regional model (Davies, 2014), this leads to biases in $CO_2$ lateral concentrations and thus affects the inversion estimates (Chen et al., 2019). The information of providing boundary conditions to regional inversions is necessary to isolate the influence of far-field contributions before performing the regional inversion. In Bayesian inversion setups, a proper information on prior error structures is also essential to determine the spatial pattern of the flux corrections based on the assumed error, especially at high spatial resolution inversions (Chevallier et al., 2012; Kountouris et al., 2015; Lauvaux et al., 2016). Therefore, the spatial pattern of flux corrections is dependent on the way the error covariance matrices are constructed, which can lead to large spatial discrepancies between the estimates from different inversion systems.

This study is dedicated to quantify the relative contributions of the differences in optimised fluxes resulting from varying: 1) atmospheric transport models, 2) lateral boundary conditions, and 3) inversion configurations on flux estimates, as the error contributions from each component to the inversions spread remain unclear in regional inversions, specifically at finer spatial scales over a continental domain such as Europe (Monteil et al., 2020; Petrescu et al., 2021; Thompson et al., 2020). We analysed results of posterior NEE estimated from the two inversion systems CarboScope-Regional, CSR, (Kountouris et al., 2018b; Munassar et al., 2022) and LUMIA (Monteil and Scholze, 2021). Both inversions employ pre-computed sensitivities of atmospheric mole fractions to surface fluxes, so-called source-weight functions or "footprints", via two Lagrangian transport models at regional scales, and make use of the two-step inversion approach established by Roedenbeck et al. (2009) to provide the lateral boundary conditions. The regional atmospheric transport models were used at a horizontal resolution of 0.25-degree. The impacts of both global and regional models were compared through analysing the differences in space and time.

Section 2 presents detailed descriptions of the inversion setups, the transport models, and the prior fluxes used. The observational stations that provide $CO_2$ dry mole fraction are described within the methods as well. We introduce the results obtained from eight inversions in Section 3. The results are discussed and interpreted through a spatial and temporal analysis of the differences between the elements of inversions in Section 4. Finally, Section 5 highlights a few concluding remarks on the impacts of regional transport, boundary conditions and inversion setups on $CO_2$ estimates in the inverse modeling.

## 2    Methods

An atmospheric tracer inversion framework is mainly made up of transport model, data source for boundary conditions (in case of regional inversions), datasets of atmospheric mole fractions, and surface flux fields. In this study, several inversion runs differing in atmospheric transport models are conducted using two tracer inversion systems, CSR and LUMIA (see

Table 2). The default CSR inversion system utilizes pre-calculated footprints from the Stochastic Time-Inverted Lagrangian Transport model STILT (Lin et al., 2003) at the regional domain, and the TM3 model at the global scale, applying the two-step scheme inversion approach (Rödenbeck et al., 2009), to provide the far-field contributions to the regional domain. In the default setup of the inversion system LUMIA, the footprints are pre-calculated using the Lagrangian particle dispersion model FLEXPART (Pisso et al., 2019), and the far-field contributions are calculated using the global transport model TM5 in a separate global inversion run, applying the two-step scheme inversion as well. These default configurations in both systems constitute the base cases. We strive to restrict the differences in the inversion runs to the targeted components, i.e., regional transport, boundary conditions, and the inversion systems, so as to outline the impact of each suite. That is, input data such as measurements of $CO_2$ dry mole fraction and the a-priori fluxes, used as constraints based on Bayes inference, are identical for all runs. We exchangeably make use of the four combinations of transport model components, the regional and global models, in the two inversion systems. The impacts were evaluated using forward model runs to quantify the differences in $CO_2$ concentrations (simulated with prior fluxes) and inversion runs to quantify the magnitude of differences in the flux space. The inversion setups and implementation are explained in the protocol of comparison (Section 2.6).

## 2.1    Inversion Framework

In the following description we remind the reader about the basic principles of the inversion schemes. For detailed information about the mathematical schemes, the reader is referred to (Rödenbeck, 2005) for CSR and to Monteil and Scholze (2021) for LUMIA. Both systems rely on the Bayesian inference that accounts for observations and prior knowledge to regularise the solution of the ill-posed inverse problem where a unique solution does not exist due to the spatial scarcity of observations. Therefore, the optimal state vector (x) is searched for in the Bayesian formalism by minimizing the cost function $J(x)$ that is typically composed of the observational constraint term $J_c(x)$ and the prior flux constraint term $J_b(x)$

$$J(x) = J_c(x) + J_b(x) \tag{1}$$

where

$$J_b(x) = \frac{1}{2}(x - x_b)^T \mathbf{B}^{-1}(x - x_b) \tag{2}$$

$$J_c(x) = \frac{1}{2}(H(x) - y)^T \mathbf{Q}^{-1}(H(x) - y) \tag{3}$$

The prior flux uncertainty defined in the covariance matrix $\mathbf{B}$ limits the departure of the control vector ($x$) to the prior flux vector ($x_b$). Similarly, the observational constraint is weighted by the observational covariance matrix $\mathbf{Q}$ that contains the so-called model-data mismatch error, including uncertainty of measurement, representativeness, and transport. This uncertainty is assigned to the diagonal of the matrix $\mathbf{Q}$ for the respective sites based on the ability of the transport model to represent the atmospheric circulation at such locations. $H(x)$ represents the atmospheric transport operator (i.e., calculated by STILT and FLEXPART in our inversions) that determines the relation between fluxes and the modeled tracer concentration, which corresponds spatially and temporally to a given vector of measurements y. Following the gradient descent method, a

variational algorithm is applied iteratively to reach the best convergence (global minimum) of the cost function that satisfies the optimal solution of the control vector. The default configurations for constructing the covariance matrices of prior uncertainty are slightly different in CSR and LUMIA. Prior flux uncertainty is assumed to be around 0.47 PgC yr$^{-1}$ over the full domain of Europe derived from the global uncertainty (2.80 PgC) assumed in the CarboScope global inversion for the annual biogenic fluxes (Rödenbeck et al., 2003). In CSR, this uncertainty is uniformly distributed spatially and temporally in a way that the annual uncertainty aggregated over the entire domain should arrive at the same value. The uncertainty structure is fit to a hyperbolic decay function in space (Eq. (4)) and to an exponential function (Eq. (5)) for the temporal decay as explained in Kountouris et al. (2015).

$$r(s) = \frac{1}{1 + \frac{s}{ds}} \qquad (4)$$

$$r(t) = e^{\frac{-t}{dt}} \qquad (5)$$

The correlation length scales $ds$ and $dt$ applied to flux uncertainties are chosen to be 66.4 km spatially and 30 days temporally, respectively, following Kountouris et al. (2018a) and Munassar et al. (2022). The spatial length in the zonal direction is set to be longer than that in the meridional direction by a factor of 2 (anisotropic), owing to larger spatial climate variability in meridional as compared to zonal direction.

The spatio-temporal shape of the prior uncertainty in LUMIA is computed in a way that each control vector comprises weekly uncertainty calculated as the standard deviation of NEE based on weekly flux variance; however, LUMIA agrees on the overall annually aggregated flux uncertainty over the entire domain with CSR. A Gaussian function of the spatial correlation decay (Eq. (6)) is applied to the prior uncertainty structure with a spatial length scale of 500 km

$$r(s) = e^{-(\frac{s}{ds})^2} \qquad (6)$$

whereas the effective temporal decay was set to 30 days (same as in CSR). Given the difference in the spatial correlation decay of the prior uncertainty, LUMIA is set to draw larger flux corrections in a broader radial area where stations exist following the gaussian decay with a longer length scale compared to the hyperbolic decay in CSR. In turn, the hyperbolic function has a larger impact in the further radial distances than the Gaussian function does, regardless of the longer spatial scale assumed with the Gaussian decay in a factor of around 7.5 in comparison with the hyperbolic decaying function.

## 2.2 Atmospheric transport models

Surface sensitivities are calculated using the STILT (Lin et al., 2003) and FLEXPART (Pisso et al., 2019) models at a horizontal resolution of 0.25-degree and hourly temporal resolution. Both models simulate the transport of air mass via releasing an ensemble of virtual particles at the locations of stations. The virtual particles are transported backward in time and driven by meteorological fields obtained from the European Centre for Medium-Range Weather Forecasts (ECMWF). STILT particles are transported 10 days backward in time and forced by forecasting data obtained from the high-resolution implementation of the Integrated Forecasting System (IFS HRES). For the FLEXPART model in standard operation,

particles are followed for 15 days backward in time driven by ERA-5 reanalysis data. To keep the consistency with STILT footprints, the backward time of FLEXPART footprints was limited to 10 days in the inversions. After this backward time integration, the particles are assumed to leave the domain, even though a large number of particles are expected to escape
after a few days. To better represent air sampling in the mixed layer, day-time observations are considered, except for mountain stations where night-time observations are used instead (Geels et al., 2007). To ensure best mixing conditions, temporal windows were considered for simulating $CO_2$ dry model fractions over stations as explained in Section 2.4 (Table 1). In addition, release heights of particles are taken as the highest sampling level above ground at each measurement site. For high altitude receptors, such as mountains, a correction height is used in STILT in a way that the actual elevation of the
station can be represented in the corresponding vertical model level (Munassar et al., 2022). In FLEXPART, the elevation above sea level is taken as the model sampling height.

## 2.3 A priori and prescribed fluxes

Three components of prior and prescribed surface-to-atmosphere fluxes of $CO_2$ are obtained from 1) biogenic terrestrial fluxes, 2) ocean fluxes, and 3) anthropogenic emissions and kept identical in both systems. Prior net terrestrial $CO_2$ exchange
fluxes, Net Ecosystem Exchange (NEE), are calculated using the diagnostic biogenic model Vegetation Photosynthesis and Respiration Model (VPRM) (Mahadevan et al., 2008). VPRM calculates NEE at hourly temporal and 0.25-degree spatial resolution, and provides a partitioning of the net flux into gross ecosystem exchange (GEE) and ecosystem respiration. Data obtained from remote sensing provided through the MODIS instrument and meteorological parameters from ECMWF drive both quantities of the light-dependent GEE and the light-independent ecosystem respiration. The model parameters were also
optimised against eddy covariance data selected within the global FLUXNET site network across Europe in 2007 (Kountouris et al., 2015). For more details on the VPRM model, the reader is referred to Mahadevan et al. (2008).

Ocean fluxes are taken from Fletcher et al. (2007), which provide climatological fluxes at a spatial resolution of 5° x 4°, remapped to 0.25-degree to be compatible with the biosphere model fluxes. In addition, anthropogenic emissions are taken from the EDGAR_v4.3 inventory, and are updated to recent years according to British Petroleum (BP) statistics of fossil fuel
consumption, and distributed spatially and temporally based on fuel type, category, and country specific emissions, using the COFFEE approach (Steinbach et al., 2011). The emissions are remapped to a 0.25° spatial grid and to an hourly temporal resolution.

Biogenic terrestrial fluxes are optimized in the inversions, while the ocean fluxes and anthropogenic emissions are prescribed, given the better knowledge about their spatial and temporal distribution in comparison with the heterogeneity,
variability, and uncertainty of the biogenic fluxes. Moreover, in the absence of observational constraints that help discriminate the contributions from the three categories, we chose to prescribe the ocean fluxes and anthropogenic $CO_2$ emissions. This is also justified by the fact that the observation sites are located in areas where the biospheric flux influence is expected to dominate the variability of $CO_2$ concentration, but it means that errors in the fossil or ocean fluxes might be compensated by the inversions, and resulting in changes in the posterior NEE.

## 2.4 Observations

Measurements of $CO_2$ dry model fractions are collected through ICOS, NOAA, and pre-ICOS stations across the domain of Europe provided by Drought 2018 Team and ICOS Atmosphere Thematic Centre (doi:10.18160/ERE9-9D85, 2020). In total, datasets from 44 stations are used covering the domain of Europe in 2018, in which a maximum number of stations is present compared to the other years. Regarding model-mismatch errors, in LUMIA a weekly value of 1.5 ppm is assumed to all sites except for the Heidelberg site where 4 ppm was assumed due to the anthropogenic influence from the neighbourhood. Table 1 denotes the weekly values of uncertainty used in CSR for the corresponding sites. The uncertainty for the surface sites is inflated to 2.5 ppm as a slight difference to LUMIA. The inflation of uncertainty from weekly to hourly values is basically calculated by multiplying weekly errors by $\sqrt{7 \times n}$ (n refers to the number of hours in the daily measurements used in the inversion). The observations are mostly assimilated as hourly continuous measurements, and are taken from the highest level, avoiding large vertical gradients near the surface that are hard to represent in the transport models. Model error in representing observations in the PBL is expected to be largest when the PBL is shallow. Therefore, for most sites, we considered data only when the PBL was expected to be well developed, i.e., during the afternoon, local time (LT). The exception is at high altitude sites, which tend to sample the free troposphere during night (Kountouris et al., 2018b). The assimilated windows are reported in Table 1.

## 2.5 Boundary conditions

Far-field contributions of $CO_2$ concentrations (originating from sources outside of the regional domain) are taken from global inversions. As default setups of the global runs, the Eulerian transport model TM3 is used in the CarboScope global inversion at 5° (lon) x 4° (lat), while TM5-4DVAR (Transport Model 5 – Four Dimensional Variational model) is used to provide boundary conditions to LUMIA using the global transport model TM5 at 6° (lon) x 4° (lat) (Babenhauserheide et al., 2015; Monteil and Scholze, 2021). Both inversion systems apply the two-step scheme inversion, explained in Roedenbeck et al. (2009), in which a global inversion is first used to estimate $CO_2$ fluxes globally (based on observations inside and outside Europe). In a second step, the global transport model is used to estimate the influence of European $CO_2$ fluxes on European $CO_2$ observations. That regional influence is then subtracted from the total concentration, to obtain a time-series of the far-field influence directly at the locations of the observation sites. This prevents introducing biases by passing concentration fields from one model to another. For detailed information about the approach methodology, the reader is referred to Roedenbeck et al. (2009).

## 2.6 Comparison protocol

The results of the study are based on eight variants of inversions differing in global and regional transport models, as well as in inversion systems as explained in Table 2. This implies, the two inversion systems (CSR and LUMIA) make use of two regional transport models (STILT and FLEXPART) and two global transport models (TM3 and TM5), which represent the

boundary conditions (background) calculated from two global inversions. Hereafter, the identifier codes (see corresponding column in Table 2) will be used to refer to the individual runs within the inversion ensemble. For instance, to highlight the impact of regional transport models, we compare the inversions that only differ in regional transport models, regardless of the inversion system or boundary conditions used, such as CS3 and CF3 or LS5 and LF5. Similarly, we use the same specifications of transport models (indicated through the identifier codes) for the forward runs to outline the differences in $CO_2$ concentrations simulated using prior fluxes with different transport models. In this case using a different system should not result in discrepancies as long as prior fluxes remain identical. In terms of system-to-system comparison, the impact of flux uncertainty should be taken into account as the prior error structure is specific for each inversion system. With that said, this has been investigated by conducting additional tests in CSR and LUMIA using identical uncertainties with flat shape and Gaussian correlation decay.

## 3    Results

Estimates of the regional biosphere-atmosphere fluxes over the domain of Europe are calculated using CSR and LUMIA for 2018 from an ensemble of eight inversions as listed in Table 2. Generally, all the inversions showed that the estimates of NEE are constrained by the atmospheric data as can be seen from the positive flux corrections made by the inversions in comparison with the prior fluxes calculated from the biosphere flux model VPRM, which obviously overestimates $CO_2$ uptake, specifically during the growing season (Fig. 1, left). This is also obvious in the ensemble-averaged annual estimates of posterior fluxes -0.29 PgC versus -1.49 PgC in the prior fluxes (Fig. 1, right). However, the spread among posterior estimates is still relatively large ranging between -0.72 and 0.20 PgC $yr^{-1}$ for the annual estimates, larger than the prior uncertainty of 0.47 PgC $yr^{-1}$. Likewise, the mean standard deviations of the monthly estimates over the ensemble of inversions is 0.72 (PgC $yr^{-1}$). The largest deviations occur between inversions that differ by the regional transport models (e.g., CS3 versus CF3, or LS5 versus LF5). In addition, the seasonal amplitude was found to be different between the STILT and FLEXPART inversions. The STILT-based inversions lead to a larger amplitude of posterior NEE than the FLEXPART-based inversions.

In terms of spatial distributions, the base cases of CSR and LUMIA inversions, i.e., CS3 and LF5 (default configurations of both systems), exhibit good agreement in predicting smaller uptake of $CO_2$ compared to the a-priori fluxes (Fig. 2, first row). The magnitude of flux corrections suggest more additional sources inferred from the atmospheric signal, as shown in the innovations of fluxes (Fig. 2, second row). Major corrections are obtained over western and southern Europe where the inversions point to an overestimation of the $CO_2$ uptake by the prior biogenic fluxes. The weak annual uptake of $CO_2$ in 2018 was exceptional and caused by the drought episode in Europe (Bastos et al., 2020; Rödenbeck et al., 2020; Thompson et al., 2020), which even turned some areas in central, northern, and western Europe into a net source of $CO_2$. The discrepancies between CS3 and LF3 noticed in the innovations, e.g., in northern France, Netherlands, and south-eastern UK are attributable to the combination of differences in regional transport models, lateral boundaries, and system configurations.

In the following we will focus on separating and quantifying the contributions of such differences caused by each driver.

### 3.1 Impact of mesoscale transport

Inversions that differ in the regional transport models (STILT and FLEXPART) demonstrate the largest differences in posterior fluxes resulting in a relative contribution of about 61% of the total differences compared to the boundary conditions and inversion systems. The differences in monthly estimates of NEE calculated with CS3 and CF3 inversion setups that vary in regional transport models are shown in Fig. 3 (top panel, "transport"). Additionally, the discrepancies caused by transport have an obvious seasonal pattern. The differences between CS3 and CF3 peak in November and June, reaching 2.11 and -

1.82 (PgC yr$^{-1}$), respectively. The best agreement between both inversions is obtained during the transitional months (August and April) with differences of -0.10 and -0.18 (PgC yr$^{-1}$), respectively. This might be attributed to the decline of the net flux magnitude during these months.

Furthermore, we assessed the impact of atmospheric transport in the simulations of $CO_2$ concentrations because this directly translates into differences in the optimised fluxes. These simulations were calculated using the total components of prior

fluxes (biosphere, ocean, and fossil fuel emissions) with STILT and FLEXPART in forward model runs to sample the atmospheric concentrations at hourly time-steps at the station locations across the site network. Note that since all runs use identical prior fluxes, it does not matter for the differences whether the prior fluxes were precise enough to reproduce the true concentration or not. Figure 3 (bottom panel, "transport") illustrates the monthly differences in the forward simulations between STILT and FLEXPART averaged over all observational stations. Similarly to the discrepancies in the optimised

fluxes, the differences in the forward simulations demonstrate a dominant impact of the regional transport model preserving the same temporal pattern as seen in the flux differences but with opposite signs. The absolute difference ranges from 0.39 to 4.37 (ppm) computed for the monthly means throughout all the sites. Geels et al. (2007) found even a larger spread up to 10 (ppm) calculated with five transport models over ten stations distributed across Europe. The notably large difference reported in that study is likely attributed to the large discrepancies in the model configurations, especially regarding the

horizontal resolution and vertical levels used. The harmonised configurations used in STILT and FLEXPART lead to a reasonably consistent representation of the atmospheric variability at synoptic and diurnal timescales. The largest differences are observed during November and May with -4.37 and 3.60 (ppm), respectively. On the other hand, the smallest differences were found to be -0.39, -0.42, and 0.56 (ppm) during September, April, and August, respectively. These results suggest a maximum impact of the mesoscale transport during the growing season and winter, while the impact converges to the

minimum during transitional months such as May and September. Overall, the differences in posterior fluxes are consistent in the timing with the differences in the simulated concentrations computed using the prior fluxes.

Further diagnostics of model-data mismatches are provided in the supplementary materials indicating the performances of STILT and FLEXPART with respect to the observations using prior and posterior fluxes across the site network at hourly, weekly and yearly time steps (see Fig. 1S and Table 1S).

In terms of the spatial discrepancies in annual flux estimates, using STILT generally leads to predicting a larger sources of $CO_2$ in the regional inversions, in particular over central Europe and the UK compared to using FLEXPART (Fig. 4, "diff: transport"). In turn, inversions using FLEXPART suggest less uptake over northern Italy, Switzerland, and south-eastern France. However, this impact refers to a spatial pattern of transport differences that might be caused either by meteorological data or by problematic sites that are hard to represent by transport models. Some areas such as north-western Italy exhibit a

persistent impact over time as shown in Fig. 4 ("sd: transport"), which shows the standard deviation of monthly differences calculated for the CS3 and CF3 inversions. In terms of temporal variations, the inversions performed with different regional transport models indicate larger monthly flux variations in comparison with those differing in global models and inversion systems (see Fig. 4, "sd: background" and "sd: system").

Figure 5 shows the spatial flux differences together with differences in prior concentrations simulated using STILT and

FLEXPART during June and December. Noteworthy, the differences in NEE, to a large extent, agree in their spatial patterns with the differences in prior concentrations calculated over the station network. In addition, there are notably particular areas that exhibit opposite signs of the spatial impact in the differences in posterior fluxes and prior concentrations such as western Europe during June and northern Europe during December. One important difference between STILT and FLEXPART is that the STILT model has higher sensitivities during summer than FLEXPART, while the opposite holds true during winter.

However, there are exceptions at individual sites such as Weybourne (WAO) in the UK and Ispra (IPR) in Italy indicating either difficult terrains that cannot be well represented by the models or real synoptic features that are resolved by one model but not by the other. The differences in forward simulations are inversely manifested in the posterior flux differences as large surface sensitivities result in smaller posterior flux corrections, and vice versa. In this case, STILT computes higher surface sensitivities than FLEXPART in June; therefore, the CS3 inversion needs to adjust less the prior fluxes to fit the

observations. On the contrary, a weaker uptake is suggested by STILT inversion during December over Europe, except for the abovementioned areas around northern Italy and south-eastern France. The differences appeared to be larger during the months of growing season and winter following the seasonal amplitude of $CO_2$.

### 3.2    Impact of lateral boundary conditions

The differences in lateral boundary conditions were found to account for about 27% of the total differences resulting from

the regional transport, lateral boundaries, and systems. This is a non-negligible contribution, albeit smaller than the regional transport contribution. The impact of using different far field contributions was analysed by assessing the differences in the posterior NEE estimated with CS3 and CS5 inversions, which use boundary conditions from the global inversions

CarboScope and TM5-4DVAR, respectively. Figure 3 ("background") shows consistent differences over time between these inversion estimates aggregated over the entire domain of Europe. Larger flux corrections are suggested by CS5 than by CS3.

This is because the global TM3-based inversion predicts higher influence at the lateral boundaries than the global TM5-based inversion does. Discrepancies in the monthly posterior fluxes between CS3 and CS5 inversions amount to a range of 0.11 to 0.64 (PgC yr$^{-1}$) absolute differences with a mean of 0.40 (PgC yr$^{-1}$). Monthly mean differences in $CO_2$ concentrations throughout all sites simulated using TM3 and TM5 boundary conditions were found to range from 0.17 to 0.93 (ppm) with a mean of 0.55 (ppm).

The distributions of spatial differences of posterior fluxes indicate a homogeneous impact across the full domain of Europe (Fig. 4, "diff: background"). Likewise, the standard deviations of the monthly posterior fluxes obtained from CS3-CS5 ("sd: background") denote flat temporal variations throughout all the grid-cells. These findings confirm the results obtained in Fig. 3 "background". This impact is consistent in space and time, with coherent deviation over all months, and is therefore expected to not affect the seasonal and interannual variability.

**3.3    Impact of inversion systems**

CS3 and LF5 differ by more than their regional transport and boundary conditions. In particular, the uncertainties are, by default, setup differently in CSR and LUMIA. The two systems optimise different set of variables (weekly NEE offsets in LUMIA and 3-horly NEE in CSR). Here we compare CS5 and LS5, which differ by their inversion systems but not by their transport model and boundary conditions. The differences in flux estimates between CS5 and LS5 inversions amount to 12%

relative to the total differences, including that caused by the mesoscale transport and lateral boundaries. This impact is, however, dependent upon system configurations, in particular the way how the prior flux uncertainty is prescribed. The absolute monthly differences between CS5 and LS5 range between 0.06 and 0.56 (PgC yr$^{-1}$) with a mean of 0.15 (PgC yr$^{-1}$) (Fig. 3, "system"). This demonstrates the smallest differences amid inversions in comparison with the transport and lateral boundary differences, which yielded absolute monthly means of 1.27 and 0.40 (PgC yr$^{-1}$), respectively. The differences

peaked during May, June, and November, while the differences remained rather small during the rest of the year. LS5 infers -6.42 and 2.39 (PgC yr$^{-1}$) during June and December, respectively, which is higher than CS5 estimates by 0.33 and 0.07 (PgC yr$^{-1}$). Generally, LS5 predicts slightly larger $CO_2$ releases compared to CS5, which is partially due to differences in how uncertainties are assumed in both systems.

The impact of uncertainty definition is quantitatively assessed through using identical uncertainties for model-data-mismatch

as well as for prior fluxes in both CSR and LUMIA. The spatial flux corrections (innovation of fluxes) shown in Fig. 8 denote quite good agreement between CSR and LUMIA estimates. In this experiment, the differences in June and December decreased to 0.23 and 0.04 PgC yr$^{-1}$, respectively, in comparison with the corresponding differences obtained from the default configurations of both systems. That is to say, the impact of uncertainty definition alone amounts to 0.09 and 0.03

PgC yr$^{-1}$ in June and December, respectively, leading to approximately 30% and 50% of the overall system-to-system
differences. The rest of the differences may be attributed to differences in the convergence of the cost function to reach the minimum values.

The spatial differences shown in Fig. 4 "diff: system" alter between positive and negative differences over the domain (but these tend to compensate when aggregating the flux estimates over the full domain). It should be noted that the inversion systems mainly differ in the definition of the shape and structure of the prior uncertainty. Therefore, applying different
structure and magnitude of prior flux uncertainty in the inversions may inflate the error in $CO_2$ flux estimates over the underlying regions in the domain, in particular if the spatial differences do not cancel out. In addition, the corresponding standard deviations of monthly estimates ("sd: system") show large temporal variations, specifically over areas that have large spatial differences. The spatial results indicate that the impact of inversion systems should not be neglected, especially at national and subnational scales.

## 4    Discussion

The regional inversions computed over Europe showed that posterior NEE is largely derived from the atmospheric signal. The seasonality of posterior NEE, inferred from the atmospheric signal, is strongly impacted by differences in the representation of atmospheric transport. Given the identical priors and observational datasets used in the inversions, using different mesoscale transport models leads to 61% of the differences in posterior fluxes in comparison with 27% and 12% of
the differences caused by the use of different boundary conditions and different inversion systems, respectively. In agreement with these results, Schuh et al. (2019) also found a large impact of mesoscale transport on estimating $CO_2$ fluxes. Hence, any error in the atmospheric transport is translated into posterior fluxes as flux corrections. For instance, CS3 and LS3 suggest annual $CO_2$ flux budgets of -0.20 and -0.72 PgC, respectively, indicating a difference of 0.51 PgC in the annual flux budget. This difference is even larger than the prior flux uncertainty (0.47 PgC). The transport also showed a large
impact on flux seasonality leading to a difference of 49% relative to the mean seasonal cycle. However, Schuh et al. (2019) found smaller differences amounting to about 10-15% of the mean seasonal cycle. Unlike the regional transport model error, the impact of boundary conditions does not show any striking seasonality and thus can be thought of as a bias in dry mole fractions. The consistency of the lateral boundary impact over time and space is in agreement with results of lateral boundary uncertainties assessed by Chen et al. (2019) using four different global transport models, albeit over a different domain.
Therefore, such an impact may be dealt with as a constant correction in mixing ratios before performing the regional inversions, potentially site-specific corrections. But there should be a reference for these corrections, for example, taking the most robust model that has been validated against observations or simply a factor of the relative mean of the relevant models/approaches. Although the inversion systems showed the smallest differences in $CO_2$ flux estimates, the specification

of the control vector (regarding the construction of covariance matrices) that devises the flux correction can result in larger
differences, specifically in the spatial flux patterns.

The large number of stations within central and western Europe lead to a strong observational constraint that is reflected in the spatial optimized fluxes over that area. Therefore, large spatial differences between the inversions are pronounced around areas where stations exist, precisely for grid cells that have non-zero footprints. The large temporal variations indicate a systematic error that possibly arises from the transport models themselves as well as from meteorological forcing data. Additionally, systematic differences between transport models occur due to discrepancies in representing vertical mixing and horizontal and vertical resolution of the models (Peylin et al., 2002). Gerbig et al. (2008) found large discrepancies in derived mixing heights between meteorological analysis from ECMWF and radiosonde data, which reached about 40% for the daytime and about 100% for the nocturnal boundary layer. The vertical mixing in tracer dispersion models was found to result in a significant variability in methane emission estimations (up to a factor of 3) given the same meteorology as investigated by Karion et al. (2019).

**Drivers of STILT-FLEXPART differences**

Although STILT and FLEXPART are run at the same spatio-temporal resolution employing similar schemes to parametrize the atmospheric motion unresolved by meteorological forcing data such as turbulence, and similar diagnostics to determine mixing heights, they still exhibit large spatial and temporal differences. A first assumption was that the differences between STILT and FLEXPART could be caused by differences in the calculation of mixing height. However, we did not find a correlation between the differences in mixing heights, calculated with the two models, and the differences in prior concentrations (Fig. 6). This finding concludes that the discrepancies in representing mixed layer heights do not explain the major differences in simulated $CO_2$ concentrations nor the differences in footprints.

The second assumption was that differences in the forcing data of meteorological products might lead to the discrepancies in both models, given that STILT uses meteorological parameters from IFS HRES, while FLEXPART uses ERA-5 reanalysis. Results in Fig. 7, "meteo", indicate that using different meteorological data results in pronounced differences when the FLEXPART model was forced by operational forecast data instead of ERA-5 reanalysis. These differences notably occur during the time of net $CO_2$ release corresponding to quite small differences during the time of growing season. This, however, only explains a small part of the overall differences (shown in Fig. 7, "base") that dominate all the months except August and September. In a previous study, Liu et al. (2011) concluded that uncertainties in meteorological fields lead to a significant contribution to the total transport error, as well as to an underestimation of the vertical turbulent mixing even when the same circulation model and mixing parameterizations were used to reconstruct vertical mixing from a single meteorological analysis. Tolk et al. (2008) also found meteorology to be a key driver of representation error, which varies spatially and temporally. They indicated that a large contribution to representation error is caused by unresolved model

topography at coarse spatial resolution during night, while convective structures, mesoscale circulations, and the variability of $CO_2$ fluxes dominate during day-time. Deng et al. (2017) found that assimilating meteorological observations such as wind speed and wind direction in transport models significantly improved the model performances achieving an uncertainty reduction of about 50% in wind speed and direction, especially when measurements in the mixed layer were assimilated.

Nonetheless, they concluded that the differences in $CO_2$ emissions reached up to 15% at local scale corrections after inversion and were limited to 5% for the total emissions integrated across the regional domain of interest. These results refer to the limited impact of meteorological data. Note however that the main aim of this experiment was to test whether differences in driving meteorological data could explain the differences between STILT and FLEXPART, but that we are not assessing the overall impact of meteorological uncertainties. Doing so would in particular require testing non-ECMWF

meteorological products.

Furthermore, we tested the possible impact of surface layer heights (the height up to which particles are sensitive to the fluxes) that may affect the particle dispersion, provided that STILT relies on the assumption of defining the surface layer as a half of the mixed layer height, while in FLEXPART it is defined as a fixed height of 100 m (these are default configurations of the models). In this experiment, STILT was run with a surface layer height of 100 m, so that the impact of the surface

layer on $CO_2$ simulations is outlined by the comparison with another run using the default configurations of STILT. The differences in simulated $CO_2$ concentrations due to differences in the surface layer were found to be quite small (Fig. 7, "s_layer") and, therefore, can be negligible in both magnitude and temporal pattern compared to the overall differences. However, varying the models STILT and FLEXPART with identical meteorological data and identical surface layer lead to the largest differences, in particular during the growing season months and winter months (Fig. 7, "model"). As a result,

model-to-model differences largely affect the simulations of $CO_2$ concentrations and are likely originating from the transport model schemes. It is clearly noticeable that the overall differences combine the underlying differences of "model", "meteo", and "s_layer", and are yielded as the arithmetic summation of this partitioning.

**How do our results explain the range of uncertainties reported in scientific literature?**

To shed more light on the drivers of differences in optimised $CO_2$ fluxes, we analyse the spread in our inversions in line with

the spreads in other inversion estimates that were reported in two previous studies over the same domain of Europe. Figure 9 shows the spreads amid the three studies: 1) eight inversions conducted in our results denoted as "Ensemble", 2) six inversions of the EUROCOM experiment "EUROCOM" done by Monteil et al. (2020), and 3) five inversions of the drought study of Thompson et al. (2020) focusing on analysing the 2018 drought impact on NEE, denoted as "Drought". Note that in EUROCOM and Drought, the tracer inversions differed in the atmospheric regional transport models, the definition of

boundary conditions, the definition of control vector, the selection of atmospheric datasets, and the a-priori fluxes. These differences are expected to span a large range of uncertainty sources in the posterior NEE. The climatological monthly

estimates of NEE were averaged over "EUROCOM" inversion members for the respective years 2006-2015, except for one inversion (NAME), which was limited to 2011-2015. "Ensemble" and "Drought" were confined to the analysis year of 2018. The monthly NEE estimates were calculated for all ensembles as the average over their respective inversion members. The annual mean of NEE estimated with "EUROCOM", "Ensemble", and "Drought" amounts to -0.19 and -0.29, and -0.05 (PgC) with standard deviations of 0.34 and 0.29, and 0.46 (PgC), respectively.

The spreads amid each ensemble of inversions are illustrated by the min and max values bounded around the mean on the error bars (Fig. 9). The monthly mean of NEE estimates shows a good consistency in all the ensembles. The spreads are also relatively comparable, albeit variable over months. For instance, "EUROCOM" and "Drought" exhibit larger spreads during the growing season (April-August), while "Ensemble" has a larger spread in the rest of months -i.e., during winter. Noteworthy, all ensembles experience large spreads during June and May. Although the participating inversions to "EUROCOM" and "Drought" had different configurations, the spreads were not largely different from our inversion spreads. This implies that the use of different atmospheric transport models could account for a large fraction of differences in posterior fluxes, although differences in the definition of uncertainty covariance matrices and lateral boundary conditions likely contribute as well. Moreover, the discrepancies in "EUROCOM" and "Drought" estimates are expected to be partially caused by using different atmospheric datasets in the inversion systems. Munassar et al. (2022) found that posterior fluxes can be more sensitive to changing the number of stations than changing the prior flux models.

## 5    Conclusions

Estimating atmospheric tracer fluxes through inverse modeling systems has been widely used, in particular targeting the major GHGs to improve the quantification of natural (both terrestrial and oceanic) sources and sinks. Here, an analysis of differences in posterior fluxes of $CO_2$ was carried out using inversion systems deploying different regional transport models. The difference between minimum and maximum spreads for annually integrated fluxes was found to be 0.92 PgC yr$^{-1}$ for the ensemble range of 0.20 and -0.72 PgC yr$^{-1}$ with a mean estimate of -0.29 PgC yr$^{-1}$ calculated over the full domain of Europe in 2018. We tested the regional transport, the boundary conditions, and the inversion systems. The regional transport accounts for the largest part of the discrepancies in the optimized fluxes as well as in the estimation of $CO_2$ concentration. Temporal and spatial differences in posterior fluxes are consistent with the differences in simulated $CO_2$ concentration sampled with STILT and FLEXPART over the station network. They demonstrate a spatial pattern over certain areas during June and December suggesting rather systematic differences between STILT and FLEXPART. The differences in the regional transport are mainly caused by the transport schemes, while meteorological forcing data partially contribute to these differences, especially during the months in which net release of $CO_2$ occurs. However, the differences in $CO_2$ simulations did not show large sensitivities to other parameters such as the way the surface layer height (maximum altitude considered sensitive to the fluxes in Lagrangian models) and the mixing height are defined. In addition, the global transport models used

in the global inversions that provide the far field contributions to the regional domain are responsible for small, but non-negligible differences in the inversion estimates. These differences appeared to be homogeneous spatially and temporally, which can be considered as bias-like. The differences arising from using different inversion systems integrated over the entire domain of Europe were on the contrary rather small, once differences such as the transport model and the uncertainties are controlled for. However, such an impact is partially a result of applying different structure and shape in the prior flux uncertainty, while the rest may be attributed to differences in the cost function convergence to reach the minimum. This reflects the importance of the way the uncertainty is prescribed in the tracer inversion systems.

The divergence in $CO_2$ flux estimates resulting from swapping the regional transport model emphasises the need for further evaluation of atmospheric transport models in order to improve the performance of the models. At the same time, it is important to realistically account for the transport errors in the tracer inversions. Errors in meteorology parameters assimilated in transport models as forcing data should also be accounted for explicitly, potentially through making use of an ensemble of meteorology data to estimate such errors. Despite the non-negligible difference between inversion systems, this study indicates the importance of following a common inversion protocol when reporting flux estimates from different inversion frameworks.

**Code and data availability**

The simulations of the ensemble of inversions (posterior NEE calculated using CSR and LUNIA), their respective prior fluxes, and codes can be made available upon request to the corresponding author. The atmospheric datasets of $CO_2$ dry mole fractions are available at the ICOS Carbon Portal and can be accessed from https://doi.org/10.18160/ERE9-9D85 (Drought 2018 Team and ICOS Atmosphere Thematic Centre, 2020).

**Competing interests**

At least one of the (co-)authors is a member of the editorial board of Atmospheric Chemistry and Physics. The peer-review process was guided by an independent editor, and the authors also have no other competing interests to declare.

**Acknowledgements**

The authors thank Mathias Göckede for his valuable comments on the manuscript in the internal review. SM, CG, CR, and F-T K, acknowledge the computational support of Deutsches Klimarechenzentrum (DKRZ) where the CSR inversion system is implemented. The computations of LUMIA were enabled by resources provided by the Swedish National Infrastructure for Computing (SNIC) at NSC, partially funded by the Swedish Research Council through grant agreement no. 2018-05973. The authors acknowledge the use of the atmospheric dataset of $CO_2$ dry mole fractions collected throughout ICOS and NOAA site network.

**Financial support**

This research has been supported by Horizon 2020 (VERIFY (grant no. 776810) and CoCO2 (grant no. 958927)).

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

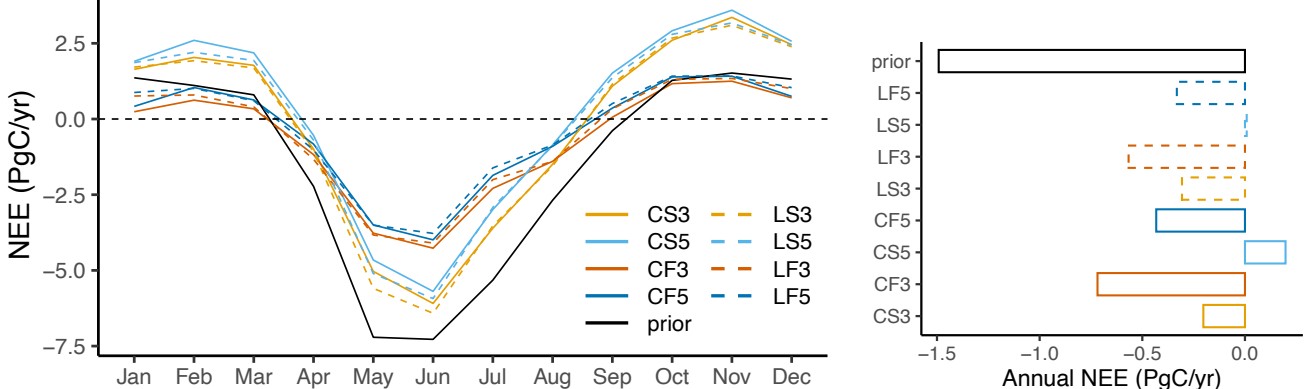

Figure 1: Left panel refers to posterior monthly NEE estimated using eight inversions, including prior NEE shown in black colour, with CSR (solid lines) and LUMIA (dashed lines), and right panel denotes the corresponding annually aggregated fluxes. Orange and red colours correspond to TM3 and dark/light blue to TM5. Orange and light blue colours refer to STILT and red and dark blue to FLEXPART.

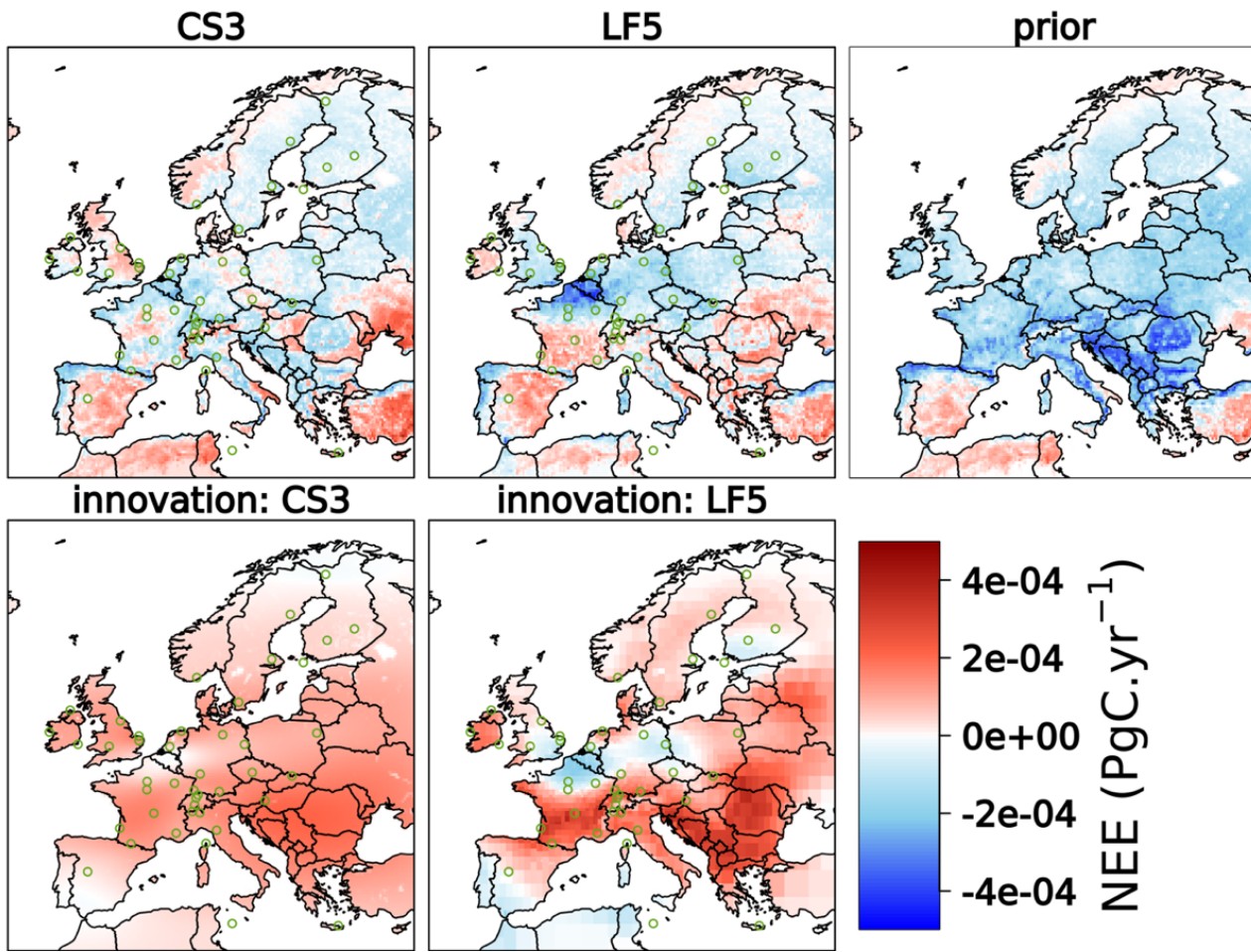

**Figure 2: First row shows the spatial distributions of annual NEE estimated with the base inversions CS3 and LF5, as well as their prior. Second row depicts the innovations of fluxes calculated for the inversions CS3, LF5. Green circles denote the locations of observational sites.**

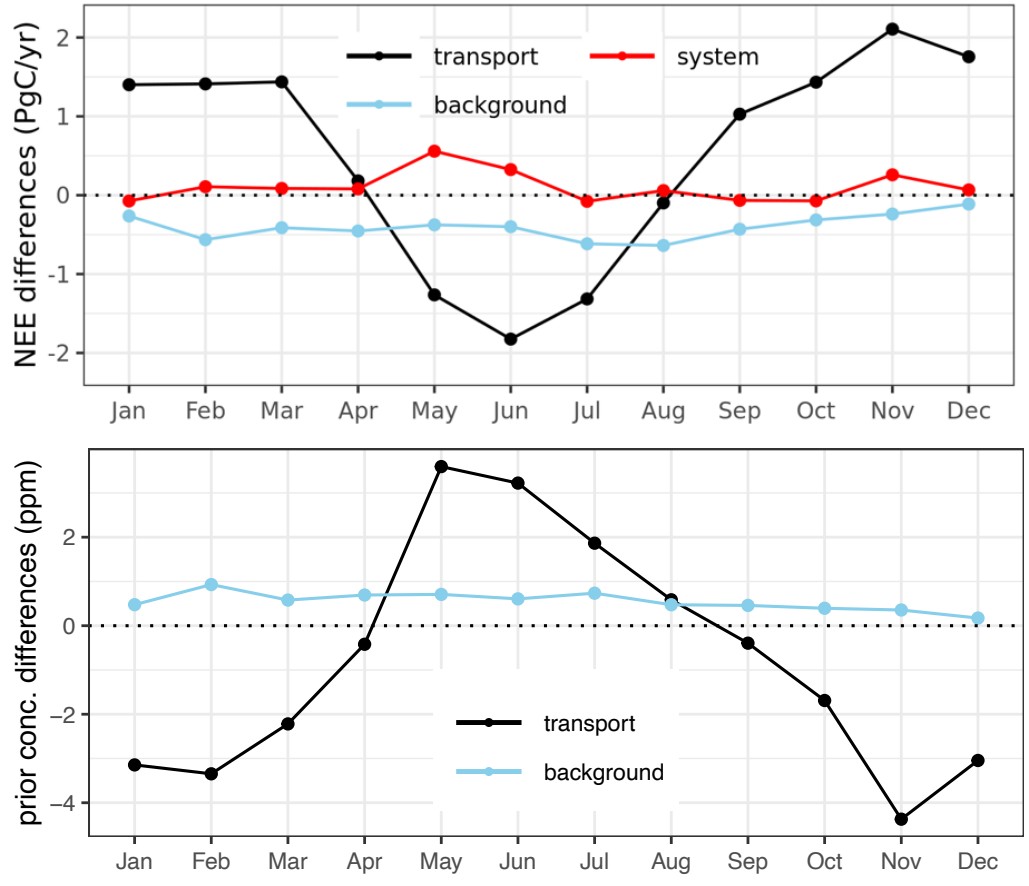

**Figure 3: Differences in optimized fluxes (top) and prior concentrations (bottom) calculated with the regional transport models STILT and FLEXPART (CS3-CF3) and background provided through TM3 and TM5 (CS3-CS5). "system" refers to the differences between CSR and LUMIA inversion for optimized fluxes (CS5-LS5).**


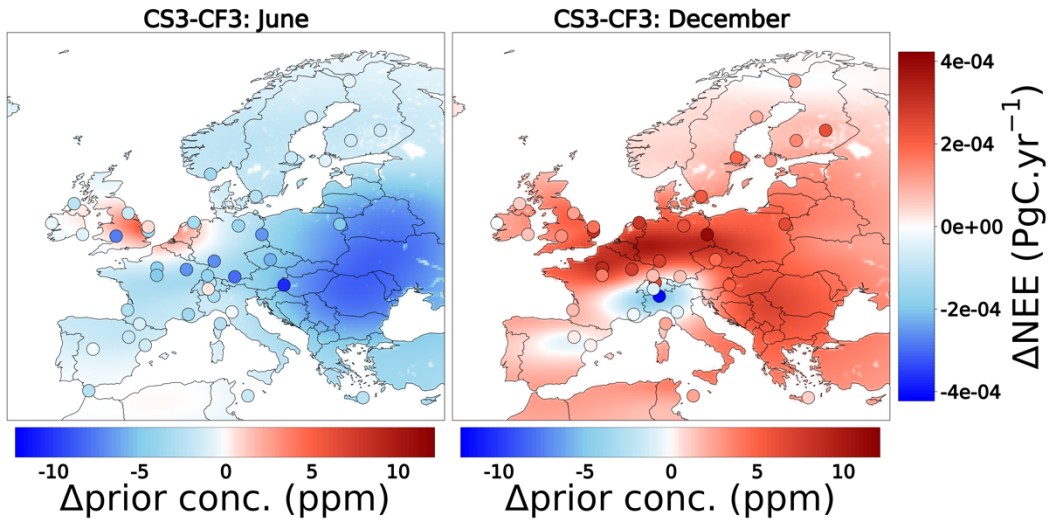

**Figure 4: First row indicates differences in annual posterior NEE estimated with STILT and FLEXPART models referred to as "transport" (CS3-CF3), TM3 and TM5 referred to as "background" (CS3-CS5), and CSR and LUMIA referred to as "system" (CF3-LF3); second row demonstrates the standard deviations of the corresponding monthly differences.**

**Figure 5: Spatial differences of posterior NEE estimated from the inversions CS3 and CF3 with STILT and FLEXPART transport models during June and December; filled circles indicate the differences in prior concentrations at the locations of sites (horizontal legend explains the magnitude of differences).**

**Figure 6: Scatter plot of differences of prior concentrations and mixing heights calculated with STILT and FLEXPART models (i.e., STILT-FLEXPART on the x- and the y-axis). Red lines indicate the slopes.**

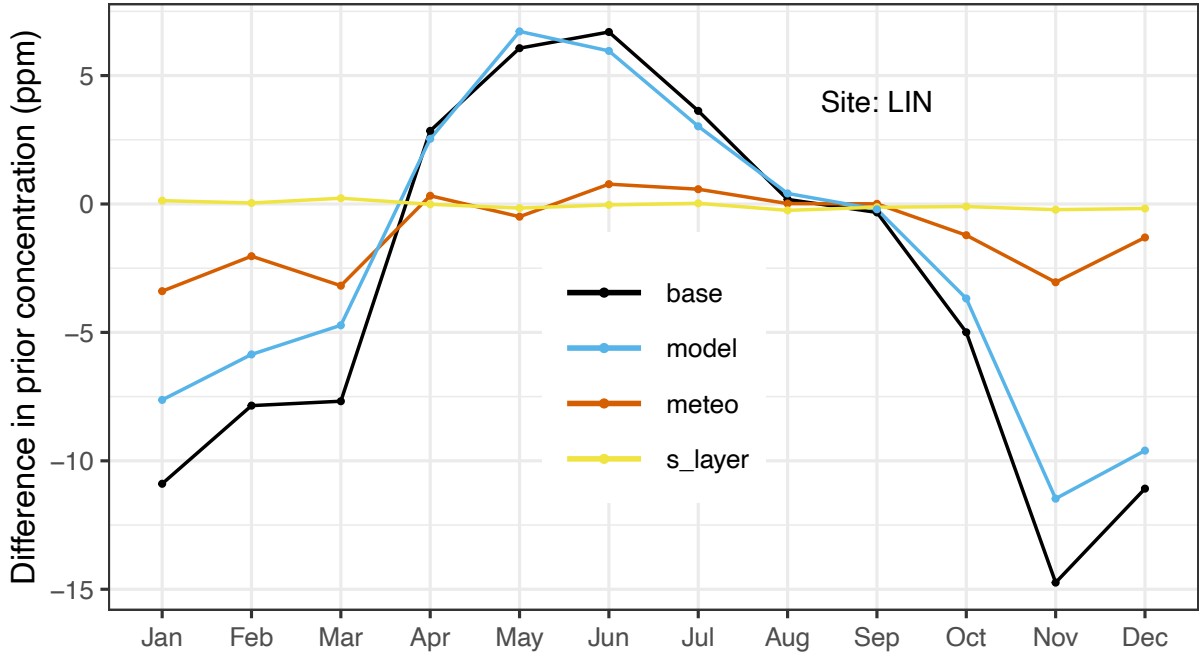

**Figure 7: Differences in prior concentration simulated at LIN with STILT and FLEXPART using different configurations.** "s_layer", yellow line, refers to the difference calculated with STILT using two assumptions of defining the surface layer height, once with the default as 0.5 of the mixed layer, and once with 100 m as used in FLEXPART. "meteo", red line, indicates the differences calculated with FLEXPART using two different types of meteorological data, IFS (the STILT default) and ERA-5. "model", blue line, denotes the differences calculated with STILT and FLEXPART, given identical meteorological data (IFS) and surface layer height (100 m). "base", black line, refers to the base configurations of STILT and FLEXPART encompassing all possible differences between models - i.e., 1) STILT with IFS forecasting data and a surface layer height as 0.5 of the mixed layer height, and 2) FLEXPART with ERA-5 reanalysis and the surface layer height of 100 m.

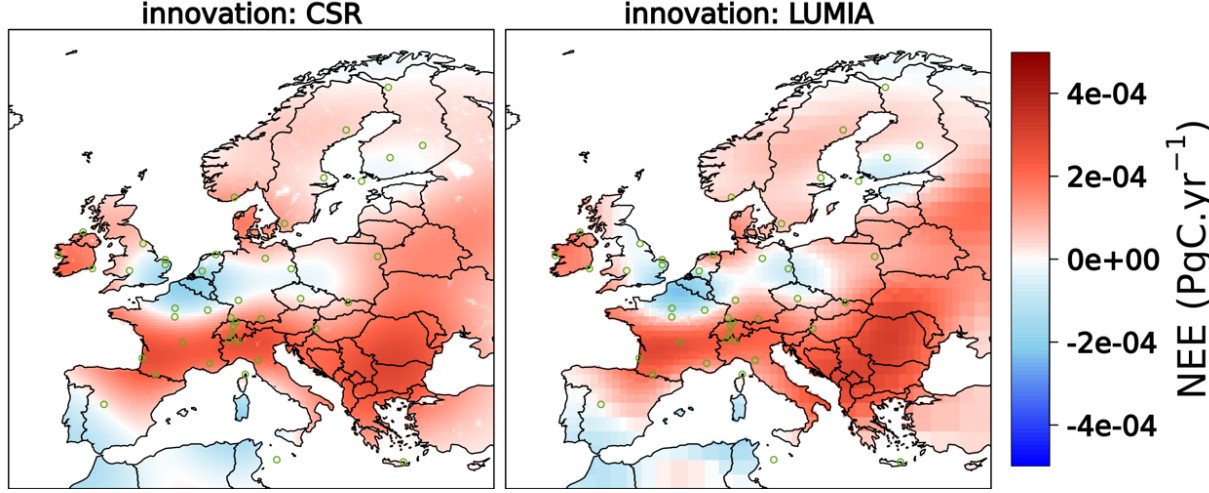

**Figure 8: Innovation of fluxes calculated from CSR and LUMIA using identical uncertainties of prior flux and measurements. The uncertainty flux shape was flat and the decaying spatial correlation was fit to Gaussian function with 500 km scale. FLEXPART and TM5 models were used in this experiment.**

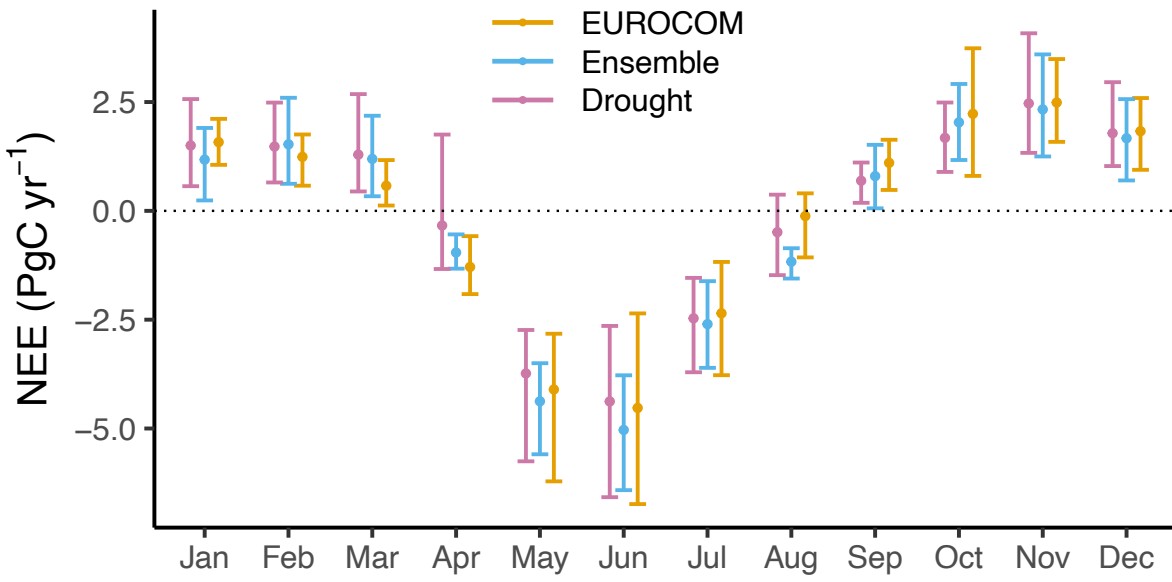

**Figure 9:** Comparison of monthly NEE estimates calculated as the mean of six inversions taken from Monteil et al. (2020), denoted as "EUROCOM", eight inversion members conducted in our study (set-ups listed in Table 2), denoted as "Ensemble", and five inversions used in Thompson et al. (2020) for the 2018 drought study denoted as "Drought". The error bars refer to the spreads (min/max) over the respective members amid each ensemble of inversions.






**Table 1. Atmospheric sites used in the inversions.**

| Site code | Site name | Coordinates (lat, lon)° | STILT release height (magl) | FLEXPART release height (magl) | Time window (UTC) | Uncertainty (ppm) |
|---|---|---|---|---|---|---|
| SM3 | Hyytiala | 61.85, 24.29 | 125 | 125 | 10:00-14:00 | 1.5 |
| BI5 | Bialystok | 53.23, 23.03 | 300 | 300 | 10:00-14:00 | 1.5 |
| FKL | Finokalia | 35.34, 25.67 | 15 | 15 | 10:00-14:00 | 1.5 |
| PAL | Pallas | 67.97, 24.12 | 12 | 12 | 10:00-14:00 | 2.5 |
| PUI | Puijo | 62.91, 27.65 | 84 | 84 | 10:00-14:00 | 1.5 |
| UTO | Uto Baltic Sea | 59.78, 21.37 | 57 | 57 | 10:00-14:00 | 1.5 |
| BIR | Birkenes Observatory | 58.389, 8.25 | 3 | 3 | 11:00-15:00 | 2.5 |
| BR5 | Beromuenster | 47.19, 8.17 | 212 | 212 | 11:00-15:00 | 1.5 |
| DEC | Deltade_lEbre | 40.74, 0.79 | 10 | 10 | 11:00-15:00 | 1.5 |
| EEC | El_Estrecho | 36.0586, -5.664 | 20 | 20 | 11:00-15:00 | 1.5 |
| GIC | Sierra de Gredos | 40.3457, -5.1755 | 20 | 20 | 11:00-15:00 | 2.5 |
| HEI | Heidelberg | 49.417, 8.674 | 30 | 30 | 11:00-15:00 | 4 |
| HP4 | Hohenpeissenberg | 47.8011, 11.0246 | 300 | 131 | 11:00-15:00 | 1.5 |
| ER2 | ERSA | 42.9692, 9.3801 | 40 | 40 | 11:00-15:00 | 1.5 |
| HT3 | Hyltemossa | 56.0969, 13.4189 | 150 | 150 | 11:00-15:00 | 1.5 |
| HU4 | Hegyhatsal Tower | 46.95, 16.65 | 115 | 115 | 11:00-15:00 | 1.5 |
| IP3 | Ispra | 45.8147, 8.636 | 100 | 100 | 11:00-15:00 | 1.5 |
| KR3 | Kresin | 49.572, 15.08 | 250 | 250 | 11:00-15:00 | 1.5 |
| LMU | La Muela | 41.5941, -1.1003 | 80 | 79 | 11:00-15:00 | 1.5 |
| LMP | Lampedusa | 35.53, 12.62 | 10 | 10 | 11:00-15:00 | 1.5 |
| LUT | Lutjewad | 53.4036, 6.3528 | 60 | 60 | 11:00-15:00 | 2.5 |
| NO3 | Norunda | 60.0864, 17.4794 | 100 | 100 | 11:00-15:00 | 1.5 |
| SV3 | Svartberget | 64.256, 19.775 | 150 | 150 | 11:00-15:00 | 1.5 |
| TR4 | Trainou | 47.9647, 2.1125 | 180 | 180 | 11:00-15:00 | 1.5 |
| OHP | Obervatoire de Haute Provence | 43.931, 5.712 | 100 | 100 | 11:00-15:00 | 1.5 |
| SA3 | Saclay | 48.7227, 2.142 | 100 | 100 | 11:00-15:00 | 1.5 |
| LHW | Laegern Hochwacht | 47.4822, 8.3973 | 400 | 32 | 11:00-15:00 | 2.5 |
| BS3 | Bilsdale | 54.359, -1.15 | 248 | 248 | 12:00 -16:00 | 1.5 |
| RG2 | Ridge Hill | 51.9976, -2.54 | 90 | 90 | 12:00 -16:00 | 1.5 |
| TA3 | Tacnolestan | 52.5177, 1.1386 | 185 | 185 | 12:00 -16:00 | 1.5 |
| WAO | Weybourne Norfolk | 52.9502, 1.1219 | 10 | 10 | 12:00 -16:00 | 1.5 |
| OP3 | OPE_ANDRA | 48.5619, 5.5036 | 120 | 120 | 14:00-17:00 | 1.5 |
| GA5 | Gartow | 53.0657, 11.4429 | 341 | 341 | 14:00-18:00 | 1.5 |
| LIN | Lindenberg | 52.1663, 14.1226 | 98 | 98 | 14:00-18:00 | 1.5 |
| BIS | Biscarrose | 44.3781, -1.2311 | 47 | 47 | 14:00-18:00 | 2.5 |
| CRP | Carnoise Point | 52.18, -6.37 | 14 | 14 | 14:00-18:00 | 1.5 |
| MHD | MaceHead | 53.3261, -9.9036 | 24 | 24 | 14:00-18:00 | 1.5 |
| MLH | Marlin Head | 55.355, -7.333 | 47 | 47 | 14:00-18:00 | 1.5 |
| JFJ | Jungfraujoch | 46.5475, 7.9851 | 720 | 3570 | 23:00-3:00 | 1.5 |
| KAS | Kasrprovy Wierch | 49.2325, 19.9818 | 480 | 1989 | 23:00-3:00 | 1.5 |
| PUY | Puy de Dome | 45.7719, 2.9658 | 400 | 1465 | 23:00-3:00 | 1.5 |

| SI2 | Schauinsland | 47.91, 7.91 | 450 | 1205 | 23:00-3:00 | 1.5 |
| PTR | Plateau Rosa Station | 45.94, 7.71 | 500 | 3480 | 23:00-3:00 | 1.5 |
| PD2 | Pic du Midi | 42.9372, 0.1411 | 1458 | 2877 | 23:00-3:00 | 1.5 |
| CMN | Monte Cimone | 44.1963, 10.6999 | 670 | 2165 | 23:00-3:00 | 1.5 |


**Table 2: List of the inversion set-ups**

| Inversion system | Transport model | Global boundary condition | Identifier code | Flux Uncertainty | |
|---|---|---|---|---|---|
| | | | | Shape | Decay |
| **LUMIA** | FLEXPART | TM5 | LF5 | Variable | Gaussian |
| **LUMIA** | FLEXPART | TM3 | LF3 | Variable | Gaussian |
| **LUMIA** | STILT | TM5 | LS5 | Variable | Gaussian |
| **LUMIA** | STILT | TM3 | LS3 | Variable | Gaussian |
| **CSR** | STILT | TM3 | CS3 | Flat | Hyperbolic |
| **CSR** | STILT | TM5 | CS5 | Flat | Hyperbolic |
| **CSR** | FLEXPART | TM3 | CF3 | Flat | Hyperbolic |
| **CSR** | FLEXPART | TM5 | CF5 | Flat | Hyperbolic |

