# Peer review of "Why do inverse models disagree? A case study with two European"

_Atmospheric Chemistry and Physics, 2022_

## Author Comment (AC1)

**Final authors' response (AC) to the referees' comments (RC1 and RC2) on acp-2022-510**

*We are very thankful to the two anonymous referees for their constructive comments. In the following, we have addressed the referees' comments under the respective sections — i.e., Anonymous Referee #1(RC1) and Anonymous Referee #2 (RC2). The manuscript has also been revised accordingly.*

*Note: The reviewer comments (RC1 and RC2) are referred to in "Arial" font type throughout the texts, and the authors' responses are referred to as "Italic Arial" with indented lines.*

**Anonymous Referee #1 (RC1)**

The authors have put together a study examining the impacts of different inverse modeling setups on estimated CO2 fluxes across Europe. I think the authors have put together a nice study that will contribute to the literature on inverse modeling and atmospheric transport uncertainties. I have a few ideas and suggestions for the authors to consider as they revise the manuscript:

**Holistic suggestions:**
(1) The manuscript text has a few English grammatical issues scattered throughout. I would do a close read for grammar before uploading the revised manuscript.

> *We went throughout the manuscript and corrected the grammatical issues, where found accordingly.*

(2) The authors test out two different choices for atmospheric models (STILT and FLEXPART), two different choices for the boundary condition, and two choices for the prior uncertainty. In several cases, the choices made in each category have some similarities. For example, both STILT and FLEXPART are driven with ECMWF meteorology (albeit different ECMWF products), and the boundary condition is generated using different versions of the TM model. I suspect that the results and uncertainties in the fluxes could look quite different if the authors had made different choices. For example, the differences due to meteorology (e.g., Fig. 7) might look very different if the authors included simulations using non-ECMWF meteorology (e.g., using a product like GEOS, MERRA-2, WRF, etc.). Similarly, differences due to the background might look very different if the authors had used a different approach, like model simulations based on GEOS-Chem or an empirical boundary condition estimate like that developed by Arlyn Andrews for CarbonTracker-Lagrange. I would be careful about generalizing the results throughout the paper and/or be vigilant about bringing in results from existing studies that may have used a wider variety of products or methodological choices.

> *We tried to restrict the differences in our experiment in transport models and in inversion systems to the minimum to realistically assess the impact of each suite on posterior estimates. That way, we calculated the differences (i.e., transport-, background- and -system-based inversions), as a metric to quantify*

*such an impact of each component and compare it with the rest. Of course, to characterise the errors, a larger number of models/ensembles would be needed to realistically represent the statistical distribution of errors. Our main focus was to address the issue of large discrepancies persistent in the results of different inversions reported recently in literature like Petrescu et al. (2021), Monteil et al. (2020), and Thompson et al. (2020), in which the modelling setups of this study were part of these published studies.*

*In terms of the lateral boundary conditions, TM3 and TM5 are different models (albeit both are Eulerian) with different parametrisations and driven by different meteorology (NCEP, ERA-5, respectively). They are used in CarboScope (TM3) and TM5-4DVAR (TM5) global inversions to provide boundary conditions through the two-step inversion approach to the regional inversions CSR and LUMIA. Both inversions used this approach using the in-situ observations available globally in both models to optimise the fluxes at global scale so as to confine the differences in boundary conditions to the global transport models as much as possible.*

*As a validation of the impact of transport models, we compared the spread over NEE estimates resulting from varying the transport models with the spread calculated with independent studies in literature (Monteil et al., (2020), and Thompson et al., (2020)). Although these studies entirely used different combinations of inversion systems with different transport models (forced by different meteo. data) and different priors, the spread in our estimates that only differ in transport models was comparable with the spreads found over the estimates of these studies as can be noticed from Fig. 9. This indicates the dominant impact of model-to-model differences that are reflected in inversion estimates apart from forcing data.*

*We agree that one should be careful about generalising the results. In this context, we only made conclusions based on our results and analysis. We carefully took the suggestion into account while revising the manuscript.*

(3) I would strongly recommend re-organizing the results and discussions sections (Sections 3 and 4). I felt that these sections often duplicated information. E.g., information presented in Sect. 3 is often repeated in Sect. 4. In addition, the text sometimes hops around from topic to topic without a strong sense of direction or flow (e.g., I felt this way about Sect. 3.1 and the beginning of Sect. 3.2.).

*We have re-organised the results and discussions sections (3 and 4) in the revised manuscript according to the suggestion.*

(a) I would specifically re-think the purpose of sub-sections 3.1 and the beginning of Sect. 3.2. These two sub-sections present information on a smattering of different topics, information that is often revisited or repeated later in Sects. 3 and 4. I think the authors intended to give a broad overview of the results in Sects. 3.1 and 3.2,

**Final authors' response (AC) to the referees' comments (RC1 and RC2) on acp-2022-510**

but the resulting text felt scattered to me and lacked a direction. For example, the authors mention forward model simulations briefly at the beginning of Sect. 3.2 and then repeat similar information about the forward simulations later in Sects. 3 and 4.

> *We agree that subsections 3.1 and 3.2 should be restructured to improve the flow and to connect it with the rest of the results. Indeed, in these subsections we opted to give an overview of the results by introducing and analysing the estimates of all inversions to facilitate the discussion of differences due to each driver of differences –i.e., subsections 3.2.1, 3.2.2, and 3.2.3.*
>
> *In the revised manuscript, we incorporated 3.1 and 3.2 into one section (under the main section 3) and rephrased text with a clear direction and also avoided the repetition, according to the suggestion.*

I would also avoid generic phrases like "large differences" or "smaller differences" when there are quantitative metrics that could be used instead.

> *Thank you for the suggestion. We avoided such generic phrases in instances in which quantitative metrics are present.*

(b) In addition, I would make sure to use strong topic sentences at the beginning of each paragraph in Sects. 3.2.1, 3.2.2, and 3.2.3. Doing so would help give each section a stronger sense of flow and direction.

> *We have done that so in the revised manuscript based on the suggestion and re-organised the three sections (now defined as 3.1, 3.2, and 3.3) for a better flow and clarity.*

(c) Furthermore, I thought it odd that Sect. 4 repeated each of the topics presented in Sects. 3.2.1, 3.2.2, and 3.2.3, often with duplicate information. I would merge the paragraphs of Sects. 3 and 4 that discuss similar topics (i.e., merge the separate discussions of transport in Sects. 3 and 4).

> *In the revised manuscript, we merged these paragraphs in the corresponding Sections of the results (i.e., Sects 3.1, 3.2, and 3.3) and skipped the duplicate information.*

**Specific suggestions:**
Line 18: "wide flux ranging between -0.72 and 0.20 PgC yr-1". Can you provide any context on how large or small this range is? I.e., what do these numbers mean to someone who isn't intimately familiar with $CO_2$ budgets? Also, what domain are you referring to here?

> *We think it is best to compare the spread to the prior flux uncertainty, so this spread is even larger than the annual prior flux uncertainty 0.47 PgC assumed over the entire domain. In addition, the spread deviates by a factor of about 1.5 from the ensemble mean. This analysis is done over the full domain of Europe. We have clarified that in the revised manuscript.*

**Final authors' response (AC) to the referees' comments (RC1 and RC2) on acp-2022-510**

Line 25 "models themselves": Can you be more precise about what you mean here?
> *The transport schemes/parametrisations of the transport models are meant here. We corrected it in the revised manuscript.*

Introduction: the introduction feels a bit like a laundry list of different papers, and some themes are mentioned multiple times at different points in the introduction (e.g., vertical mixing and PBL heights). There are definitely a lot of papers in the literature on atmospheric transport errors. It could be more effective to organize the different paragraphs of the introduction around specific themes. E.g., have one paragraph focused on vertical mixing, another on advection, etc.
> *In the revised manuscript the introduction has been reorganised based on the suggestion, taking these points into account.*

Introduction: What knowledge gaps are there in the existing literature that this study attempts to fill? The introduction doesn't really address this question at present; instead the authors frame this study as yet another study on top of the existing studies they mention.
> *The main focus of this study is to quantify the impact on atmospheric inversion estimates resulting from the choice of transport, boundary conditions and the inversion setups. Indeed, there are many studies in literature laid out in investigating such impacts but with different designs. For instance, The TransCom intercomparison (Gurney et al., 2002, 2004, and 2016) and Peylin et al. (2002) were dedicated to discussing the sensitivities of inversions to transport models at global scales with rather coarse resolution that cannot determine the real atmospheric variability; Deng et al. (2017) and Chen et al., (2019) tested the impact of assimilating ensemble of/different meteorological data in the atmospheric transport model WRF; Lin and Gerbig (2005) evaluated the transport error by incorporating uncertainties in wind fields into stochastic motions of air parcels. Karion et al. (2019) compared estimations of methane emissions at local source point using different meteorology and dispersion models and found dispersion models to be responsible for a large variability in the estimates by a factor of 3 (this supports our findings).*
> *The goal of our study is laid out in disentangling the issues of large spread that are still persistent over different regional inversions participating in different projects done recently such as EUROCOM and VERIFY, e.g., (Monteil et al. 2020; Petrescu et al. 2021; Thompson et al. 2020) devoted to estimate GHG fluxes at the domain of Europe, given the dense atmospheric observations from which inversions benefit towards robust data-driven estimates and operationality. The transport models (STILT, FLEXPART, TM3, and TM5) and the inversion systems CSR and LUMIA were participating in these studies for those projects. As a result, the outcome of this experiment has quantitatively shown the magnitude of differences that can be taken into account in reasonably prescribing the atmospheric errors in the inversions, but also in*

**Final authors' response (AC) to the referees' comments (RC1 and RC2) on acp-2022-510**

> *developing the atmospheric transport model parameterisations themselves (apart from the meteorological impact) to better reach good convergence in inversion results.*

Introduction: I recommend taking a look at Karion et al. (2019, https://acp.copernicus.org/articles/19/2561/2019/). That study is focused on methane, not CO2, but I think it might be relevant for the themes in the present manuscript.

> *Thank you for this recommendation. It is really relevant for our study and supporting our results. It shows significant differences between dispersion transport models (Lagrangian) in estimating methane emissions given the same meteorology, albeit the study was confined to a site level. We included some citations in the revised manuscript.*

Lines 179-180: How were these model-data mismatch errors chosen?

> *They were chosen based on the ability of atmospheric transport models to represent the atmospheric circulation at the station locations. Therefore, the measurements have different errors based on the station type, e.g., tower, mountain, ocean, etc. as has been explained by Rödenbeck (2005). Nonetheless, the model-data mismatch errors do not affect our analysis as they were held identical for each suite of comparison.*

Line 211 "However" -- I don't think this is quite the word you want here. Maybe "With that said ...."

> *It is corrected in the manuscript according to the suggestion.*

Lines 240-245: Do you have any numbers or quantitative metrics that show the relative impacts of these different model parameters on the fluxes? At present, it's not entirely clear what kind of differences the text is referring to.

> *The impacts were found to be 61%, 27%, and 12% relative to the total differences for the mesoscale transport, lateral boundaries, and system, respectively. We added these details in the beginning of the results Sections (3.1, 3.2, 3.3) as we re-organised the sections of results and discussions (3 and 4) based on the suggestion in "Holistic suggestions".*

Line 250 "quite large differences": Can you be more specific?

> *This has been clarified in Section 3.1 in the revised version (see the reply to the previous comment).*

Section 3.2.2: Differences in background estimates yield smaller uncertainties in the posterior fluxes relative to differences in atmospheric transport models. I wonder if that result could be due, in part, to the fact that both background estimates are generated using relatively similar models (e.g., TM3 vs TM5).

**Final authors' response (AC) to the referees' comments (RC1 and RC2) on acp-2022-510**

> *TM3 and TM5 are different models (albeit both are Eulerian) with different parametrisations and forcing data (TM3 forced by NCEP and TM5 by ERA-5). In this experiment we tried to restrict the differences to the global transport models TM3 and TM5 only, given that they don't result in large $CO_2$ biases in comparison with other BC products such as CAMS. Therefore, the same method of the two-step inversion used to calculate the far field contributions is applied, as well as we mostly used the same in-situ observations available across the globe to optimise the fluxes at the global scales. Of course, the differences are smaller and more consistent than those found in the regional transport models but they are not negligible as they lead to around third of the total differences in the optimised fluxes. Applying a different approach or product of boundary conditions to provide BCs might result in different results but still prone to larger biases.*

Lines 326-332: Most of this information repeats information provided at the beginning of Sect. 3.
> *This has been rephrased in the revised version. Additionally, Sect. 3 is restructured based on the suggestion.*

Lines 332-342: This material feels like it belongs in 3.2.1. I.e., this information on atmospheric transport seems like it belongs in the sub-section on atmospheric transport.
> *The material has been moved to the result Sect. 3.1 accordingly.*

Lines 349-350: This is the third or fourth location where the manuscript mentions forward runs. I would either (1) Consolidate all results/discussion of the forward runs within a single section, or (2) Divide this discussion into the sub-sections of Sect. 3 corresponding to different model parameters.
> *We agree, this is now consolidated into the results, Sect. 3.1 where forward simulations presented and discussed in the revised manuscript.*

Lines 3540-364: This text feels like it belongs in Sect. 3.2.1 with the results on atmospheric transport.
> *We moved this text into Sect. 3.1 accordingly in the revised manuscript.*

Lines 368-370: I wasn't able to follow this sentence.
> *This sentence is rephrased in the revised manuscript.*

Lines 402-425: The text feels like it belongs in Sect. 3.2.2.
> *It has been incorporated into Sect. 3.2 in the revised version. The text is also readapted with the context of that subsection.*

Lines 407-409: I'm not sure that I follow here. Who would apply this correction and in

what circumstances?

> *Such corrections can be done after calculating the boundary conditions in the first step of the inversion (that is, before performing the regional run). But there should be a reference for these corrections, for example taking the most robust model that has been validated against observations or simply a factor of the relative mean of the relevant models/approaches. Then one can decide what is the best way of applying these bias corrections either at a site level or applying one correction over all the sites.*
> *This has been clarified in the revised version.*

Lines 413-425: This text appears to repeat information in Sect. 3.2.3. I think the text in this paragraph belongs in Sect. 3.2.3.

> *In the revised manuscript, we dropped the repetitive information and consolidated the rest of the material into Sect. 3.3 based on the suggestion.*

Figure 7: I suspect the differences due to meteorology could be much larger if using a non-ECMWF product. E.g., if using GEOS, MERRA-2, WRF, etc.

> *We agree, using such different meteorological products might lead to larger differences than we got. Doing that would need a further study that can be designed to assimilate variety of meteo. data in one transport model, which is not technically available in our transport modelling setup. However, in this study the main focus is given to address the divergence in inverse modeling to estimate CO2 fluxes at regional scales. The interesting result was that although we used identical forcing data, the transport scheme itself leads to large differences up to around 11 (ppm) in the regional signal. Such differences are not trivial as they also translate into large discrepancies in estimating the fluxes in the inversions. As a result, this study sheds more light on analysing and quantifying this type of impact for calling for further studies on model developments and better accounting for in the inversions.*

**Anonymous Referee #2 (RC2)**

In this study, the authors have examined the leading drivers of CO2 flux inversion by setting up an ensemble of eight inversions with different regional transport (STILT and FLEXPART), global transport (TM3 and TM5), and inversion systems (CSR and LUMIA) over Europe in 2018. The surface measurement of CO2 dry mole fraction is used to estimate the posterior. The results exhibit a large spread of CO2 flux estimation, majorly driven by the mesoscale transport model, rather than two other contributions. Overall, this study is interesting and worthy of investigation; however, some queries need to be addressed.

**Major comments:**

I don't think the title could well represent the entire manuscript. This study quantified

**Final authors' response (AC) to the referees' comments (RC1 and RC2) on acp-2022-510**

the impacts of regional transport, lateral boundary conditions, and inversion systems on estimating biogenic CO2 flux. Even though the major contribution of regional transport was found and more results laid out in order to dig into details in context; still, the main part of the study is not only about atmospheric transport. Otherwise, you may want to use more regional transport models and focus majorly on the impact of atmospheric transport.

> *We agree with the suggestion that the title should represent the entire manuscript despite the focus on atmospheric transport. In the revised manuscript, the title has now changed to "Why do inverse models disagree? A case study with two European $CO_2$ inversions" accordingly.*

Similar to 1), I recommend re-organizing the introduction part as well in order to give a better bridge to readers. The current introduction is mostly focusing on atmospheric transport (Line 30 – 57).

> *In the revised manuscript, we have re-organised the introduction based on the recommendation.*

The number of ensemble members for each suite representing mesoscale transport, global transport, and inversion system is too limited (two of each) with overlapped characteristics. For instance, TM3 and TM5 should have similar characteristics. Also, I doubt the result of differences of "meteo" shown in Figure 7. Those are all the ECMWF datasets. It would be better to use ECMWF and non-ECMWF to present the impact of using different types of meteorological data.

> *We agree the ensemble members are limited but are still representative to assess the impact by analysing differences for each suite as we did not assess the error that would need enough samples and a different study design. Both TM3 and TM5 are different models with different parameterisations and forced by different meteorology (NCEP and ERA-5) as well. FLEXPART and STILT have different mechanisms in resolving turbulence and vertical mixing, although they are both Lagrangian. All these models are widely used in inverse modeling, and thus important to look into the cause of their differences. Interestingly, model-to-model differences in Fig. 7 showed large discrepancies between models given the identical meteorology, which is worthy to look into in order to bridge the gaps between models for better convergence in the invers modeling estimates. We mainly aimed at digging more into the causes of large spreads seen over regional inversion estimates reported recently in literature –e.g., Monteil et al., (2020), Thompson et al., (2020), and Petrescu et al., (2021), given that the modeling setups in this study were part of these studies.*

Line 181-182: Why do two inversion systems use different uncertainty for the observation?

> *Only 6 sites out of 44 that have a slight difference of 1 (ppm) (see Table 1) – i.e., the sites that were classified as surface stations have 2.5 ppm in CSR,*

> *while in LUMIA such sites were treated same as towers. Nevertheless, we used identical uncertainty in both systems for the experimental setup of system-to-system evaluation. Other than that, it has nothing to do with the differences in transport and boundary conditions as they were evaluated in one system.*

**Minor comments:**

Line 25: "the largest impact seems to come from the models themselves" is too vague.

> *It has been clarified in the revised manuscript.*

Line 46 – 53: The accuracy is the meteorological data is not addressed in the main text.

> *This has been addressed in the revised version as the introduction is reorganised.*

Equations 1-3: It's better to add equations containing a sensitivity term (calculated by STILT and FLEXPART)

> *The equations represent the cost function the inversions minimise containing the term of atmospheric sensitivities (i.e., the Jacobian matrix H(x) in Eq. 3) calculated by both STILT and FLEXPART models. We clarified that in the respective text.*

Line 228 – 231: From figure 2, major corrections appeared over western and southern Europe, not around the observational sites.

> *We rectified that in the revised manuscript based on the correction.*

---

## Referee Report (RR1)

Review – Why do inverse models disagree?  A case study with two European CO2 inversion by Munassar et al.

The authors addressed the comments well, and I can see many improvements throughout the manuscript. I have minor suggestions that encourage the authors to consider and discuss in the manuscript before publication.

1. Line 428 – 452: It is not clear what the author is intended to address by showing the comparison between model output with two different studies.  This study aims to find the dominant drivers of amid spread of CO2 estimates, but there are a large number of differences in between this study and two different studies by Monteil et al. (2020) and Tompson et al. (2020). Therefore, comparing ranges of CO2 inversion from this study and others is not enough to claim as "(Line 448) This implies that the use of difference atmospheric transport models could account for a large fraction of differences in posterior fluxes" so it needs to provide more results.

2. I still encourage the authors to discuss the potential uncertainties/bias by using a limited number of ensemble members in CO2 inversion in the discussion.